# Selective Dehydration of Pentoses and Hexoses of *Ulva rigida* to Platform Chemicals Using Nb_2_O_5_ and ZrO_2_ Supported on Mesoporous Silicas as Heterogeneous Catalysts

**DOI:** 10.3390/ijms262010054

**Published:** 2025-10-15

**Authors:** Gabriela Rodríguez-Carballo, Benjamín Torres-Olea, Cristina García-Sancho, Julia Vega, Félix L. Figueroa, Juan Antonio Cecilia, Pedro Maireles-Torres, Ramón Moreno-Tost

**Affiliations:** 1Departamento de Química Inorgánica, Cristalografía y Mineralogía, Facultad de Ciencias, Universidad de Málaga, Campus Universitario de Teatinos s/n, 29071 Málaga, Spain; gabrielarc@uma.es (G.R.-C.); benjamin@uma.es (B.T.-O.); jacecilia@uma.es (J.A.C.); maireles@uma.es (P.M.-T.); rmtost@uma.es (R.M.-T.); 2Facultad de Ciencias, Instituto Universitario de Materiales y Nanotecnología (IMANA), Universidad de Málaga, Campus Universitario de Teatinos s/n, 29071 Málaga, Spain; 3Facultad de Ciencias, Instituto Interuniversitario de Investigación en Biorrefinerías (I3B), Universidad de Málaga, Campus Universitario de Teatinos s/n, 29071 Málaga, Spain; 4Departamento de Ecología, Facultad de Ciencias, Instituto Universitario de Biotecnología y Desarrollo Azul (IBYDA), Universidad de Málaga, Campus Universitario de Teatinos s/n, 29071 Málaga, Spain; juliavega@uma.es (J.V.); felixlfigueroa@uma.es (F.L.F.)

**Keywords:** *Ulva rigida*, dehydration of monosaccharides, Nb_2_O_5_ and ZrO_2_ supported on mesoporous silicas, furfural, 5-hydroxymethylfurfural

## Abstract

Furfural and 5-hydroxymethylfurfural are considered as essential platform molecules for the chemical industry, acting as precursors and intermediates of numerous products. They are produced from pentoses and hexoses, respectively, in an acid medium. In this work, biomass from a green macroalgae, *Ulva rigida*, was treated under acidic conditions provided by heterogeneous catalysts in order to promote the dehydration of its monosaccharides into furfural and 5-hydroxymethylfurfural. Particularly, two functionalized mesoporous silicas, HMS and SBA-supported metal oxides (Nb_2_O_5_ and ZrO_2_), were used as catalysts. Their textural, structural, and acid properties were deeply studied, providing excellent BET surface areas (ranging 424 to 1204 m^2^/g) and a high concentration of acid sites (220–460 µmol/g), which then translated into great catalytic performances (77.8% and 64.1% of furfural and HMF molar yields, respectively, using HMS-Nb) after a 4 h of reaction time at 180 and 160 °C, respectively. The catalyst showed excellent stability and recyclability as it could be reused for up to five reaction runs with only a slight decrease in performance.

## 1. Introduction

Biomass represents an abundant, accessible, eco-friendly, and cost-effective carbon source [1] and can substitute traditional fossil resources in many ways for the manufacture of products of very different natures [2] including those that are generally obtained from fossil sources such as fuels [3,4] and chemicals [5].

The exploitation of fertile land and human labor has led to a competitive situation between food crops and crops used as biomass feedstock for bioproducts and biofuels [6,7]. For this reason, there has been a shift over the past years toward using non-edible biomass from lignocellulosic origin [8,9] in order to avoid this competition. However, when it comes to its valorization, lignocellulosic biomass presents a considerable drawback, as the presence of lignin in its composition hinders the effective extraction of its main polysaccharides [10,11]. Therefore, biomass of algal origin has become an interesting alternative [12,13] for bioethanol and platform chemical production from carbohydrate-rich macroalgae [14,15]. The usage of biomass from algae has many other advantages; not only can they be grown artificially in an easy and fast way, but also their recovery from beaches and lakes [16,17] where they can proliferate in massive blooms due to eutrophication [18] or the lack of a natural consumer, and is highly beneficial, as this proliferation can jeopardize marine ecosystem dynamics and fishing activities [19]. Macroalgae are subdivided into three main categories depending on the main pigment they present: red algae (*Rhodophyta*), brown algae (*Phaeophyta*), and green algae (*Chlorophyta*) [20]. Green algae are the most similar to terrestrial plants, as chlorophyll is their main pigment and they are rich in cellulose and hemicellulose as well as their land analogues. The *Ulva* genus is one of the most common of green algae genres. *Ulva* spp. (e.g., *Ulva rigida*), also known as sea lettuce, is a species that is able to grow in different environments relatively easily, and it is considered an invasive alga in China, which even disrupted the marine activities of the Beijing Olympics in 2008 due to the massive green tides spotted in the Yellow Sea due to its uncontrolled proliferation [21,22]. However, it not only affects seawater, but also freshwater, as large algal blooms can appear in lakes and ponds due to eutrophication [23,24]. For this reason, affected areas have begun to develop strategies for its valorization. Due to its considerably high content in carbohydrates, *Ulva rigida* has raised attention as a possible feedstock for the production of bioethanol [25] as, in addition to cellulose and hemicellulose, which, as previously said, are common to all green macroalgae, they are also composed of a sulfated polysaccharide named ulvan, composed of structural blocks of xylose, sulfated rhamnose, iduronic acid, and glucuronic acid [26,27]. Carbohydrates are also an interesting source that can be used as feedstock for the production of platform molecules, such as furfural [28] and 5-hydroxymethylfurfural (HMF) [29], two furan-derived molecules that can act as intermediates for the synthesis of a great variety of products. These two analogues can be easily obtained from the dehydration of the monomeric units of the polysaccharides present in the alga.

Both hydrolysis and dehydration reactions require the presence of acid sites. Although mineral acids are traditionally employed, there is a trend to replace them by heterogeneous catalysts. HMS and SBA silicas are commonly used when the target reaction requires a great surface area. These SiO_2_ materials present a considerable contribution of medium-sized pores or mesopores to the porous structure [30,31], which provides better access to the active centers and a better diffusion of products and reactants. They are excellent supports for metals and metal oxides [32,33], which make them ideal for the preparation of catalysts with different properties depending on the supported species. The dehydration of sugars to furans requires the presence of an acidic medium, which can be provided by a solid catalyst, such as a metal oxide, which can be supported on HMS or SBA silicas by means of simple methods such as wet impregnation. As previously mentioned, metal oxides can be easily incorporated, such as Nb and Zr, into silica using the conventional wet impregnation method. These oxides have acidic properties and present both Lewis and Brønsted acid sites with different strengths, which are ideal to carry out the dehydration of pentoses and hexoses. Both of these metals have previously been used in dehydration reactions in the literature as active phases of different catalysts such as Nb_2_O_5_-TiO_2_ [30] for the dehydration of fructose to HMF or Nb_2_O_5_-ZrO_2_ for the hydrolysis and dehydration of cellulose [31], demonstrating their excellent properties for the promotion of the catalytic dehydration of sugar and derivatives.

In this work, the exploitation of *Ulva rigida* as a novel third-generation biomass residue, a green macroalgae responsible for notorious green tides all over the world, was studied as biomass feedstock for the catalytic production of furfural and HMF by replicating the monosaccharide composition (glucose, rhamnose, and xylose) of the liquor resulting from the hydrolysis of its main polysaccharides: hemicellulose, cellulose, and ulvan [32] as a prior optimization to the utilization of the liquor of *Ulva rigida*. The catalytic acid sites necessary for the dehydration to take place were provided by metal oxides (Nb_2_O_5_ and ZrO_2_) supported on mesoporous silicas (HMS and SBA). The effect of each support and the presence of either one or both metals in the catalyst on the overall performance of the synthesized catalysts was further studied. The material that presented the best performance was also retested in several reaction cycles, and its stability and activity was investigated as well as a comparison of its catalytic performance by using commercial carbohydrates and a real liquor from *Ulva Rigida* macroalgae.

## 2. Results and Discussion

### 2.1. Characterization of Catalysts

#### 2.1.1. N_2_ Sorption at −196 °C

N_2_ adsorption-desorption isotherms at −196 °C were employed for the determination of the textural properties of the materials. According to the BET surface area results (Table 1), the HMS silicas possessed, in general, a superior surface area compared with those of the SBA silicas, with HMS-Nb being the catalyst that presented the highest surface area of 1204 m^2^/g. Nevertheless, the surface area of all silicas was remarkably high. All of the synthesized catalysts (Figure 1) presented a Type IV isotherm [33], showing a limited hysteresis loop for the HMS-based catalysts and H4 type in the case of the SBA-based ones, which generally reflects a small quantity of mesoporous limited by micropores [34].

Alongside the shape of the isotherms and the BET surface area, other porosity features were also determined (Table 1). The t-plot micropore area provided valuable insights into the estimated micropore contribution to the porosity of the materials. In general, the HMS materials had the highest t-plot areas (679–975 m^2^/g), and therefore the highest micropore to external surface area ratios. However, when it came to the overall pore volume, the HMS materials (0.54–0.73 cm^3^/g) generally exceeded the SBA silicas (0.23–0.42 cm^3^/g). Despite the differences in the pore volumes, the average pore diameter was similar in all cases, as the relation of surface to pore volume was different for each material.

The pore size distribution analysis (Figure 2) revealed that, in general, the SBA-based catalysts presented a more heterogeneous distribution of pore sizes, as they showed two contributions, one with a lower pore width (around 20 Å) and another with higher values (around 40 Å), with SBA-Nb being the one that presented the highest second contribution at approximately 45 Å. The HMS catalysts showed a narrow pore size distribution, centered at around 35 Å. The SBA silicas presented two contributions to the pore size distribution, as one of them was attributed to the mesoporous structure and the other one to the micropores associated with the silica channels and connections between pores [35,36], while the HMS silicas showed a very ordered and uniform mesoporous structure due to their synthesis through the generation of micelles, and therefore a narrow distribution of their pores [37].

#### 2.1.2. X-Ray Diffraction

The X-ray diffraction patterns revealed the amorphous character of materials and the successful dispersion of metal oxides over the supports (Figure 3), with the exception of SBA-Zr, which presented some low intensity peaks at 2θ: 30, 50, and 60°, attributed to the (111), (220), and (311) planes, respectively [38], due to the presence of some ZrO_2_ particles.

Only the SBA-based catalysts showed a signal between 1 and 1.5°, which can be associated with the (100) reflection of the long-range hexagonal arrangement of SBA materials [39] (Figure 3A–C). The HMS-based materials (Figure 3D–F) did not present any signals in the small-angle range.

#### 2.1.3. X-Ray Photoelectron Spectroscopy

The superficial composition (Table 2) and oxidation state of the chemical species of the catalysts were analyzed by means of X-ray photoelectron spectroscopy (XPS) (Appendix A).

As expected, since the supports selected for the synthesis of these catalysts were silicas, the surface was indeed composed of Si and O, in a Si/O ratio slightly lower than the theoretical expected ratio (0.5). The extra O present on the silica could be attributed to the oxygen being part of the supported metal oxides. The theoretical molar ratio of Si/M that we intended to achieve was 5, as pointed out in Section 3. In general, the superficial ratio tended to approach the desired value for the monometallic oxides. However, the materials that were impregnated with both metal precursors presented a Si/(Nb + Zr) surface ratio far from 5. This difference could be associated with the fact that the impregnation was carried out in two steps: first impregnating the support with the Nb precursor, and then with the Zr precursor, leading to coverage of the silica surface and more niobium and zirconium species on the surface.

The oxidation state of the Nb and the Zr was also possible to determine through XPS (Appendix A, Figure 4 and Figure 5). Irrespective of the support, in the catalysts containing Nb_2_O_5_, the only species found was Nb^5+^, as the Nb 3*d* doublet at 206 and 210 eV confirmed [40,41], and in the materials impregnated with zirconium, it showed the characteristic Zr 3*d* doublet at 183 and 185 eV [42,43], corresponding to Zr^4+^ in an oxidic environment.

#### 2.1.4. Transmission Electron Microscopy and Energy Dispersive X-Ray Spectroscopy

The morphology of the materials was studied by means of TEM analysis (Figure 6). Moreover, in order to confirm the dispersion of the metal oxides over the surface of the silicas, an EDX analysis was carried out. Data reported in the literature on the morphology of HMS mesoporous materials have already suggested the presence of globular structures [44]. The HMS-based catalysts showed particles with a globular structure; these globular structures are the results of the kind of synthesis carried out for their obtention, where the surfactant along with the template and the solvent create a micellar-like structure that then collapses and evolves into HMS when the silica precursor is added. In the case of SBA-based catalysts, the characteristic hexagonal parallel channels were clearly discerned in the micrographs of the catalysts, the result of encapsulating the structural director molecule or template into the silica framework creating a rigid and robust support. When the micrographs of the synthesized catalysts were compared with those of the pristine supports (Appendix A), it was observed that the deposition of the metal oxides caused no considerable change on the structure of the supports, as the HMS maintained its globular shape while SBA kept the microchannels unchanged. The EDX analysis of both the HMS- and SBA-based catalysts showed a uniform dispersion of Nb or Zr, as was also inferred from the XRD data, as no relevant crystallographic signals were observed (Figure 3), with the exception of SBA-Zr (see Appendix A), thus discarding the formation of segregated oxide species of both metals (Appendix A).

#### 2.1.5. Ammonia Thermoprogrammed Desorption

The total acidity was measured via ammonia thermoprogrammed desorption (Figure 7), and the results of the integration of curves are presented in Table 3.

All desorption curves presented a broad band with discernible shoulders. This fact shows a heterogeneity of acid sites with different strengths for all of the synthesized catalysts. The bimetallic catalysts showed the highest concentration of acid sites, and those based on monometallic Zr had the lowest acidity values (Table 3).

On the other hand, the HMS support provided a higher concentration of acid sites than the SBA support; it is likely that this support better disperses the active phase and therefore more active sites are available. However, when the density of acid sites was considered, the best results were those from the SBA-based catalysts, since these catalysts showed lower surface areas. These results, as summarized in Table 3, show that the majority of the acid sites present in these materials were of weak (100–200 °C) and moderate (200–300 °C) nature [45], with a limited contribution of strong acid sites (300–550 °C). The ratio of weak to moderate nature tended toward 2/1 for most silicas, with the exception of HMS-NbZr and HMS-Nb, which showed a 1/1 ratio. The total acidity was in general similar, although the Nb_2_O_5_ catalysts manifested a slightly higher acidity when compared with the ZrO_2_ analogues. The highest acidity was achieved by the SBA-NbZr catalyst (460 μmol/g).

#### 2.1.6. Pyridine Adsorption Coupled to FTIR Spectroscopy

The Lewis and Brønsted nature of the acid centers of the materials was studied via pyridine adsorption coupled to FTIR spectroscopy (Figure 8).

The absorption band located at ca. 1450 cm^−1^ was assigned to the 19*b* vibration mode of pyridine interacting with Lewis acid sites, whereas that at ca. 1545 cm^−1^ was associated with the 19*b* vibration mode of the pyridinium cation, resulting from the interaction of the pyridine molecule with Brønsted acid sites. The band at approximately 1490 cm^−1^ corresponded to the 19*b* vibration mode of pyridine, which refers to the mixed interaction of the pyridine with both Lewis and Brønsted acid sites [46,47,48]. Through the integration of these signals, the acidity of the materials can be obtained, as proposed by Emeis [49]. The temperature of desorption of the pyridine can be correlated with the acid strength in such a way that pyridine molecules desorbed at 100, 200, and higher than 300 °C are associated with weak, moderate, and strong acid strength, respectively.

It can be inferred from Figure 8 that the Nb-based catalysts had a greater concentration of both L and B acid sites than the Zr-based ones. Moreover, the bimetallic catalysts showed the highest concentration of L and B acid sites; therefore, some synergy between Zr and Nb oxides should occur for these catalysts. This behavior agrees well with the NH_3_-TPD results, where the bimetallic catalysts showed the highest total acidity.

In general, the trends resulting from the acid analysis by NH_3_-TPD were also valid for pyridine adsorption (Table 4), with the exception of HMS-Nb, which presented a significant difference in the weak to moderate acid site ratio. In this sense, while the weak to moderate acid site ratio of HMS-Nb according to NH_3_-TPD was close to 1/1, the value obtained from pyridine adsorption was approximately 3/1. This difference can be attributed to the fact that pyridine is a larger molecule than NH_3_, and therefore, there are acid sites that are more difficult for pyridine molecules to access due to steric hindrance since they cannot diffuse through the whole porous structure of the materials. This is also the reason why the acidity values obtained from pyridine adsorption were lower than those deduced by NH_3_-TPD, as ammonia is a much smaller molecule that can easily diffuse through narrow cavities and reach more acid sites.

Nevertheless, HMS-Nb still exhibited a considerable high acidity, especially Lewis acidity, whose value was the highest out of all the characterized materials. In general, all catalysts containing only Nb_2_O_5_ presented a higher number of Lewis acid sites compared with Brønsted ones, the vast majority being of both weak and moderate strength. These Lewis acid sites are commonly associated with very distorted octahedral species of [NbO_6_], whereas the Brønsted acid sites are usually related to these species but are much less distorted [50,51,52]. The silicas containing ZrO_2_ exhibited a different trend, as in general, they presented a nearly 1/1 ratio of Brønsted to Lewis acid sites, even when Nb_2_O_5_ was also present on the material. The species responsible for the acid nature of the sites of the materials containing Zr were unsaturated Zr^4+^ for Lewis acidity and hydroxyl groups adjacent to the former for Brønsted acidity. The catalyst that presented the highest overall acidity through pyridine adsorption was HMS-NbZr (162 μmol/g), which was distributed into a 2/1 ratio of Brønsted to Lewis acid sites. This material also presented the highest number of Brønsted acid sites (59 μmol/g), while the highest Lewis acidity (198 μmol/g) was shown by HMS-Nb.

### 2.2. Catalytic Results

The autohydrolysis of *Ulva rigida* resulted in a hydrolysate composed mainly of rhamnose, xylose, and glucose (Appendix A), which seems consistent with the fact that *Ulva rigida* is a green macroalgae formed by three structural polysaccharides: ulvan, hemicellulose, and cellulose. Moreover, total hydrolysis was performed on the hydrolysate in order to determine the potential quantity of monosaccharides that the *Ulva rigida* could yield. In order to optimize the conditions for the production of HMF and furfural, commercial rhamnose, xylose, and glucose were used in the same proportions as those determined from the total acid hydrolysis of the hydrolysate (Appendix A). Thus, the reaction temperature and time were evaluated for the dehydration of pentoses and hexoses at the same time. At 180 °C, the production of HMF from hexoses was favored (Figure 9A) over the dehydration of the pentoses toward furfural (maximum yield 2.7% at this temperature), as depicted in Figure 9B, as furfural is highly sensitive to its polymerization and the formation of other by-products at high temperatures, so lower temperatures are more suitable for this furan-derived molecule to coexist with HMF [53].

The SBA-Zr presented very low activity at both reaction temperatures of 160 and 180 °C for the dehydration of the sugars into HMF and furfural, presumably due to the fact that this material had the lowest acidity, as confirmed by NH_3_-TPD and pyridine adsorption, and the worst textural properties and dispersion of the active phase, as suggested through XRD. For HMS-NbZr, despite its excellent acid properties (see Table 3 and Table 4) and a great surface area and pore volume, its performance for both temperatures was lower than expected. This could be attributed to the fact that HMS-NbZr presented the lowest external surface and the highest micropore to external surface area ratio, meaning that the diffusion of sugars and products could be hindered, especially for hexoses and HMF, as they are larger than pentoses and furfural. Its bimetallic analogue, SBA-NbZr, presented a very different behavior. Although their acid properties were similar and its surface area was significantly lower than that of HMS-NbZr, SBA-NbZr possessed a greater external surface area and a much lower micropore to external surface area ratio due to the porous structure composing this silica, which was rather mesoporous as its pore size distribution confirms (see Figure 2). This feature was especially beneficial for the production of HMF, and it translated into an excellent performance of the catalyst. However, even if SBA-NbZr seemed superior in terms of acidity to SBA-Nb, it still outperformed the former in the production of furfural. This could be related to the higher surface area of SBA-Nb, and especially to the higher external surface area and pore volume. The only catalyst that outperformed SBA-Nb out of the catalysts assayed was HMS-Nb for both HMF and furfural production. HMS-Nb presented the highest BET surface area, and although there was a high contribution of micropores, the external surface area was considerably high as well as the pore volume and diameter, which were the highest out of the synthesized catalysts. In terms of acidity, HMS-Nb possessed one of the highest values according to NH_3_-TPD and pyridine adsorption as well as the highest number of Lewis acid sites, a necessary condition for the isomerization of glucose into fructose [54], and for the dehydration of the latter into HMF [55], along with a fair number of Brønsted acid sites, responsible for the dehydration of pentose into furfural [56]. All of these characteristics resulted in the highest catalytic performance out of all silicas. Surprisingly, its Zr analogue did not show the same potential for HMF production; nevertheless, this can be easily justified, as HMS-Zr had a lower BET surface area and a considerably smaller average pore diameter, along with a much lower Lewis acidity. However, unlike SBA-Zr, HMS-Zr was acid enough in terms of Brønsted acidity to promote the dehydration, but the average pore diameter was probably responsible for a slower diffusion, and thus the peak performance of this catalyst was achieved after 6 h of reaction time. The Nb_2_O_5_-based catalysts presented a higher acidity than those based on ZrO_2_, and in general, when Nb was present, the number of Brønsted acid sites was also higher [57,58,59]. In general, when the active phase was Nb_2_O_5_, the reaction yields were higher, but the deactivation of the catalyst was also faster, diminishing the reaction yield after 4 h of reaction time. A reason for this fast deactivation compared with the Zr-based catalysts could be found when studying the acidity through pyridine adsorption coupled to FTIR spectroscopy. Generally, the Nb-containing catalysts presented a higher number of weak Lewis acid sites that could allegedly be more active than the rest of the catalytic sites but more prone to deactivation. The pristine supports were also assayed in the catalytic dehydration of the sugars. As expected, the yields provided by the pristine silicas were significantly lower than those that presented the deposition of metal oxides such as Nb_2_O_5_ and ZrO_2_.

The reusability of the best catalyst, HMS-Nb, was evaluated for the production of both furfural and HMF under the experimental conditions that led to the best catalytic results (160 °C–4 h and 180 °C–4 h, respectively) (Figure 10).

The catalyst robustly carried out the dehydration of the sugars without barely any loss of performance during the first three and four cycles when the target molecules were furfural and HMF, respectively, even if there was no treatment of regeneration between cycles. In both cases, after the fourth catalytic run, there was a noticeable diminishment in the yield of both furans. This decrease could have been caused by the large quantity of micropores in HMS-Nb, which are more easily blocked by the adsorption of products resulting from secondary reactions after several catalytic runs.

After confirming that HMS-Nb could be successfully reused for the conversion of commercial sugars into platform molecules (furfural and HMF), it was assayed in the dehydration of a real solution obtained by the hydrolysis of *Ulva rigida* (Figure 11A). A very similar catalytic performance to the replicated liquor with commercial sugars, expressed in terms of efficiency for both furfural and HMF, was attained, demonstrating that even after involving other components in the liquor that could be present in real biomass, the catalyst was still active and performed excellently under both experimental conditions, as similar tendencies and close reaction yields could be achieved. When comparing the synthesized catalyst to the other catalysts prepared with commercial fumed silica (Figure 11B), the improvement in the production of both products was evident, partly due to the better deposition of Nb_2_O_5_ on the supports and partly due to the mesoporous contribution to the porosity and the higher BET surface area, as the commercial fumed silica used presented a BET surface area of 200 to 300 m^2^/g according to the supplier.

## 3. Materials and Methods

### 3.1. Materials

Rhamnose, tetraethylorthosilicate (TEOS) (98%), ZrOCl_2_·8H_2_O (98%), oxalic acid (99%), and Pluronic-123 (80%) were purchased from Sigma-Aldrich (Sigma-Aldrich, St. Louis, MO, USA), whereas methyl isobutyl ketone (MIBK), H_2_SO_4_ (95%), and glucose (99.8%) were acquired from VWR (VWR, Radnor, PA, USA). Dodecylamine (97%) and Nb_2_(C_2_O_4_)_5_ were purchased from Thermo Fisher Scientific (Waltham, MA, USA). Xylose (98%) was acquired from Millipore (Burlington, MA, USA). Fumed silica was purchased from Sigma Aldrich.

The macroalga *Ulva rigida* was recovered from the coast of Málaga (36°42′43″ N, 4°19′19″ W), Andalusia, Spain. The detailed identification and pretreatment of the macroalga can be found in the Appendix A.

### 3.2. Synthesis and Characterization of Catalysts

The mesoporous silicas synthesized in this work (HMS and SBA) were prepared according to previously published methods [39,60]. The silicas acted as the support of the metal oxides, Nb_2_O_5_ and ZrO_2_, which were incorporated into their surface by means of wet impregnation using Nb_2_(C_2_O_4_)_5_ and ZrOCl_2_ as precursors. The catalysts were obtained after calcination at 550 °C and presented a Si/M molar ratio (M = Nb or Zr) of 5. When working with the mixed metal catalyst, the molar ratio was still Si/M = 5, and the moieties of precursors used for the single impregnations were halved. The catalysts were denoted as *X*-*y*, where *X* indicates the support employed (HMS or SBA) and *y* symbolizes the metal oxide used, expressed as Nb, Zr, or NbZr if both were utilized. The detailed synthesis can be found in the Appendix A.

All of the synthesized materials were characterized by means of several techniques. The porous surface area was measured by means of N_2_ sorption at −196 °C using an ASAP 2420 by Micromeritics (Micromeritics, Norcross, GA, USA) after degasifying the sample at 200 °C for 24 h. The structure was studied by means of X-ray diffraction (XRD) in θ-θ mode on a PANalytical EMPYREAN diffractometer (Bruker, Rheinstetten, Germany). The long-range structural order was determined by means of small-angle X-ray scattering in a Bruker D8 Discover. The overall morphology of the catalysts was analyzed via transmission electron microscopy (TEM) in a Talos F200X by Thermo Fischer Scientific (Thermo Fisher Scientific, Waltham, MA, USA) at 200 kV and 10.5 nA. The chemical composition of the surface was analyzed by means of X-ray photoelectron spectroscopy (XPS) with a Mg-radiation PHI5700 spectrometer (Physical Electronics, Eden Prairie, MN, USA) and an Al K_α_-radiation PHI Versa Probe II spectrometer, with a constant energy pass of 29.35 eV and a beam diameter of 720 and 100 μm, respectively. The acid properties of the catalysts were determined via ammonia thermoprogrammed desorption (NH_3_-TPD) in an Autochem lll by Micromeritics (Micromeritics, Norcross, GA, USA) by saturating NH_3_ (10% He) at 100 °C for 15 min after undergoing a cleaning step at 550 °C for 30 min at a 10 °C/min heating rate. The measurements were registered between 60 and 550 °C at a 30 °C/min rate for 15 min. The discrimination between Brønsted and Lewis acid sites was possible by means of pyridine adsorption coupled to FTIR spectroscopy in a Tensor 27 spectrometer by Bruker (Billerica, MA, USA), the number of Brønsted and Lewis acid sites were calculated by integrating the bands appearing at 1545 and 1455 cm^−1^, and applying the molar extinction coefficients of 1.67 and 2.22 cm/µmol, respectively.

### 3.3. Catalytic Tests

#### 3.3.1. Dehydration of Monosaccharides

The dehydration assays were carried out in 15 mL Ace pressure^®^ (Sigma Aldrich, St. Louis, MO, USA) reactors by adding 30 mg of solid acid catalyst along with methyl isobutyl ketone (MIBK) and water (1/3 volume ratio) and commercial rhamnose, xylose, and glucose, replicating the potential concentrations in the macroalgae hydrolysate after carrying out the total hydrolysis at 100 °C for 1 h in 4 wt.% H_2_SO_4_. After the set reaction times, the phases were separated, the organic phase was diluted with methanol in an 8/1000 ratio, and then both phases were filtered over 0.45 μm filters.

#### 3.3.2. Catalyst Reutilization

After a catalytic run, the rest of the solution after the preparation of the sample was carefully decanted and discarded. Then, the catalyst, which remained at the bottom of the reactor, was washed repeatedly with 10 mL of acetone until reaching a colorless solution. Finally, the catalyst was left to dry before reusing it in another catalytic cycle with fresh reaction solution.

#### 3.3.3. Analysis Conditions of the Reaction Products

The reaction was followed by means of HPLC (AS-2055 (autoinjector), PU-2089 (pump), RI-2031 PLUS (RI detector), MD-2015 (UV detector), co-2065 (furnace), and JASCO, Easton, MD, USA). The aqueous phase of each catalytic trial was analyzed using 0.0025 M H_2_SO_4_ set at 0.35 mL/min as the mobile phase and a Phenomenex (Torrence, CA, USA) Rezex ROA-Organic Acid H^+^ (8%) (300 × 7.8 mm) column thermostated at 40 °C as the stationary phase. The organic phase was evaluated using methanol as the mobile phase flowing at 0.5 mL/min using a Phenomenex LUNA 5 μm C18 (250 × 4.6 mm) column at room temperature.

#### 3.3.4. Furfural and HMF Quantification

The yield of furfural and HMF obtained after the catalytic assays was expressed in terms of molar quantities as in Equations (1) and (2), respectively, for the case of the commercial mixture, and as efficiency in Equations (3) and (4) for the dehydration of the *Ulva rigida* hydrolysate, defining efficiency as the quotient of the quantity of target molecule in mol and the potential quantity of pentoses or hexoses in mol. The potential pentoses or hexoses were calculated by carrying out an acid hydrolysis on the *Ulva rigida*, as further described in the Appendix A.(1)$$Molar\;Yield\;\left(\%\right)=\frac{Mol\;Furfural\;}{Mol\;pentoses}\cdot 100$$(2)$$Molar\;Yield\;\left(\%\right)=\frac{Mol\;HMF}{Mol\;hexoses}\cdot 100$$(3)$$Efficiency\;\left(\%\right)=\frac{Mol\;Furfural\;}{Mol\;Potential\;pentoses}\cdot 100$$(4)$$Efficiency\;\left(\%\right)=\frac{Mol\;HMF}{\;Potential\;hexoses}\cdot 100$$

## 4. Conclusions

This work constitutes a preliminary approximation of the catalytic production of furfural and HMF from algal biomass in the presence of monometallic and bimetallic Nb_2_O_5_ and/or ZrO_2_ supported over HMS and SBA silica, whose textural (BET: 421–975 m^2^/g), acidic (TPD: 220–460 μmol/g), and structural (globular and hexagonal honeycomb like structure) were analyzed in depth. The dehydration of the sugars mainly presents in *Ulva rigida* hydrolysates (rhamnose, xylose, and glucose) was successfully carried out, attaining excellent molar yields. Moreover, the tuning of the experimental conditions made it possible to achieve a selective dehydration of the pentoses and hexoses that depended only on the reaction temperature. HMS-Nb was found to be a suitable catalyst for the dehydration of both pentoses and hexoses into furfural (77.8%) and HMF (64.1%), respectively. This catalyst was also tested in several catalytic runs and maintained the same yield up to four catalytic runs without any pretreatment, thus proving to be a robust material. After assaying the reutilization of the catalyst, we finally decided to test it with a real liquor obtained from *Ulva rigida* macroalgae, providing considerably high efficiency results (68.4% for furfural production and 54.6% for HMF production) that were comparable to the results obtained when the liquor was replicated using commercial sugars.

## Figures and Tables

**Figure 1 ijms-26-10054-f001:**
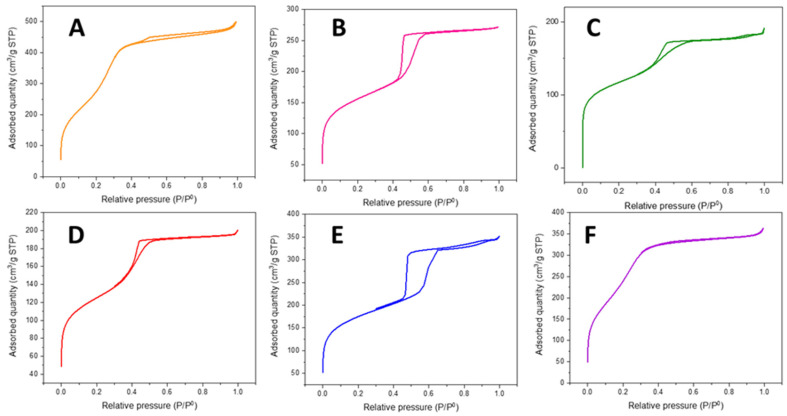
N_2_ adsorption-desorption isotherms of (**A**) HMS-NbZr, (**B**) HMS-Nb, (**C**) HMS-Zr, (**D**) SBA-NbZr, (**E**) SBA-Nb, and (**F**) SBA-Zr.

**Figure 2 ijms-26-10054-f002:**
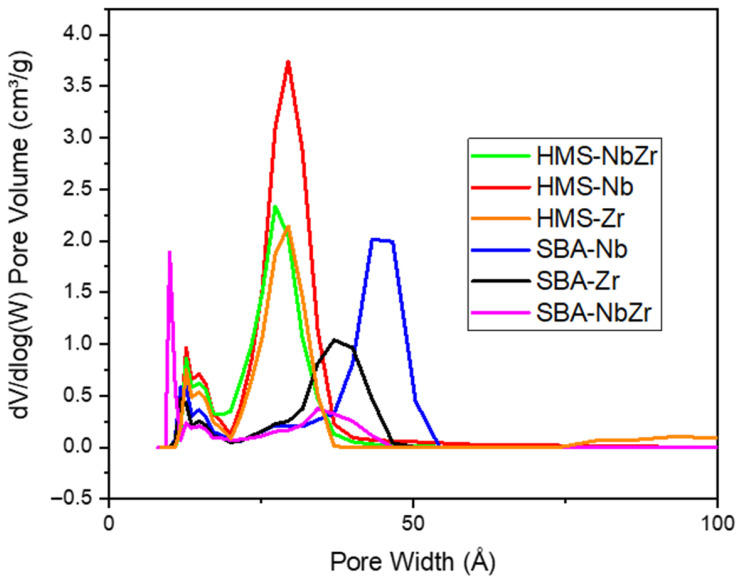
Pore size distribution of the synthesized catalysts.

**Figure 3 ijms-26-10054-f003:**
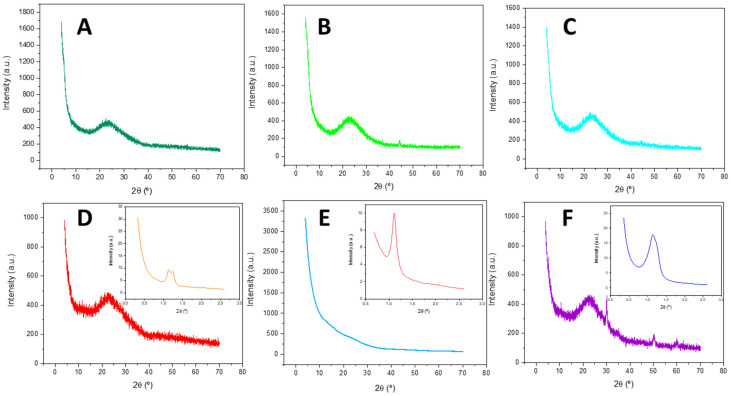
X-ray diffraction patterns of (**A**) HMS-NbZr, (**B**) HMS-Nb, (**C**) HMS-Zr, (**D**) SBA-NbZr (inset: small-angle X-ray scattering pattern of SBA-NbZr), (**E**) SBA-Nb (inset: inset: small-angle X-ray scattering pattern of SBA-Nb), and (**F**) SBA-Zr (inset: inset: small-angle X-ray scattering pattern of SBA-Zr).

**Figure 4 ijms-26-10054-f004:**
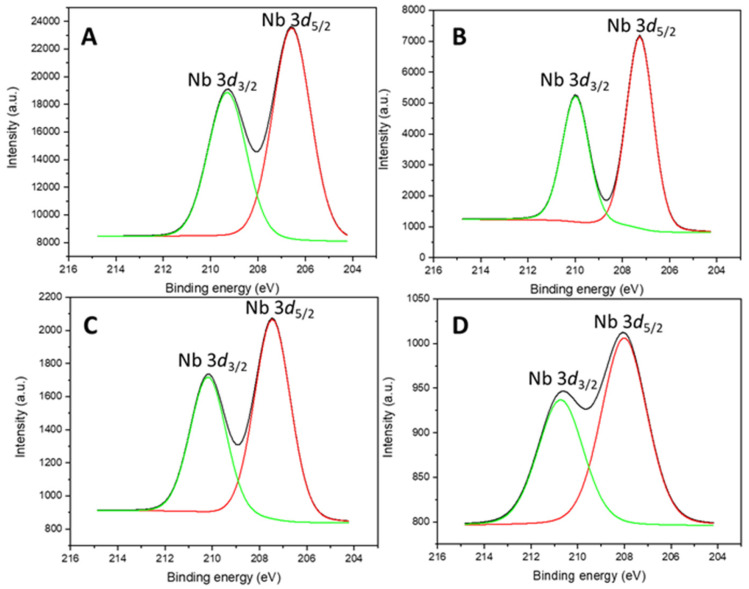
Nb 3*d* core level spectra of (**A**) SBA-Nb, (**B**) SBA-NbZr, (**C**) HMS-NbZr, and (**D**) HMS-Nb.

**Figure 5 ijms-26-10054-f005:**
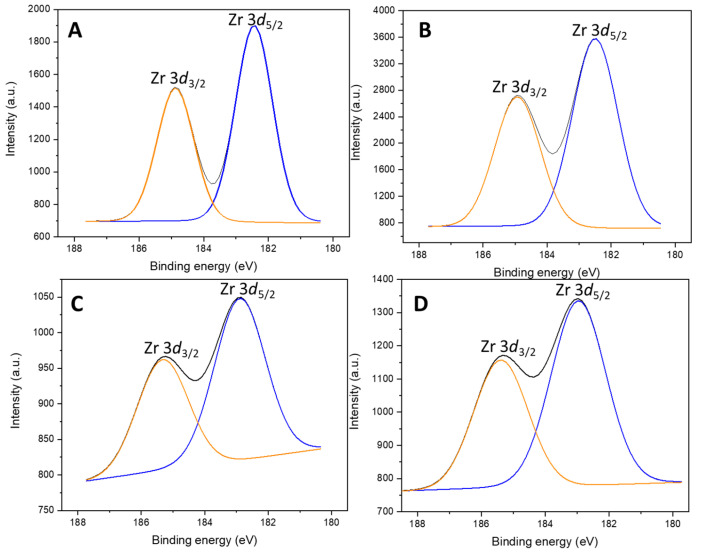
Zr 3*d* core level spectra of (**A**) SBA-Zr, (**B**) SBA-NbZr, (**C**) HMS-NbZr, and (**D**) HMS-Zr.

**Figure 6 ijms-26-10054-f006:**
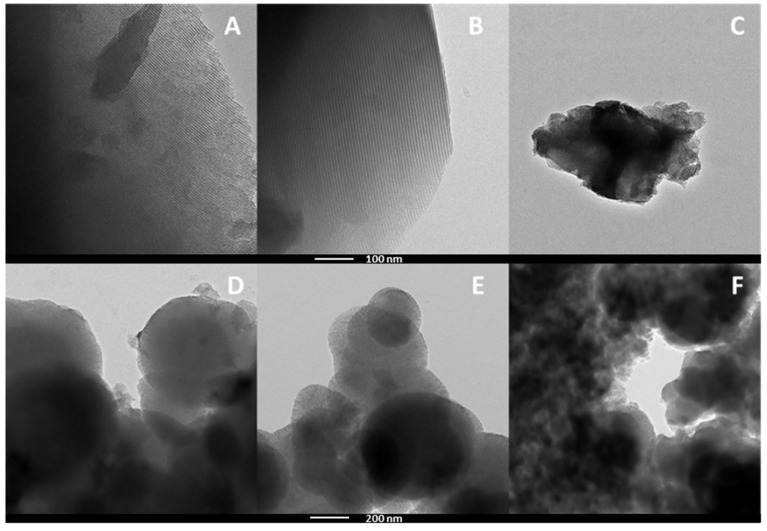
TEM micrographs of (**A**) SBA-NbZr, (**B**) SBA-Nb, (**C**) SBA-Zr, (**D**) HMS-NbZr, (**E**) HMS-Nb, and (**F**) HMS-Zr.

**Figure 7 ijms-26-10054-f007:**
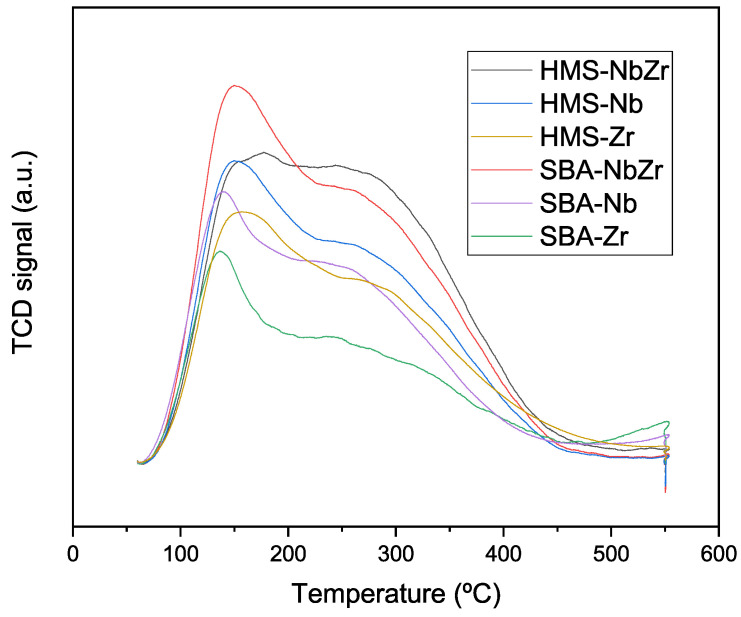
NH_3_–TPD curves of the synthesized catalysts.

**Figure 8 ijms-26-10054-f008:**
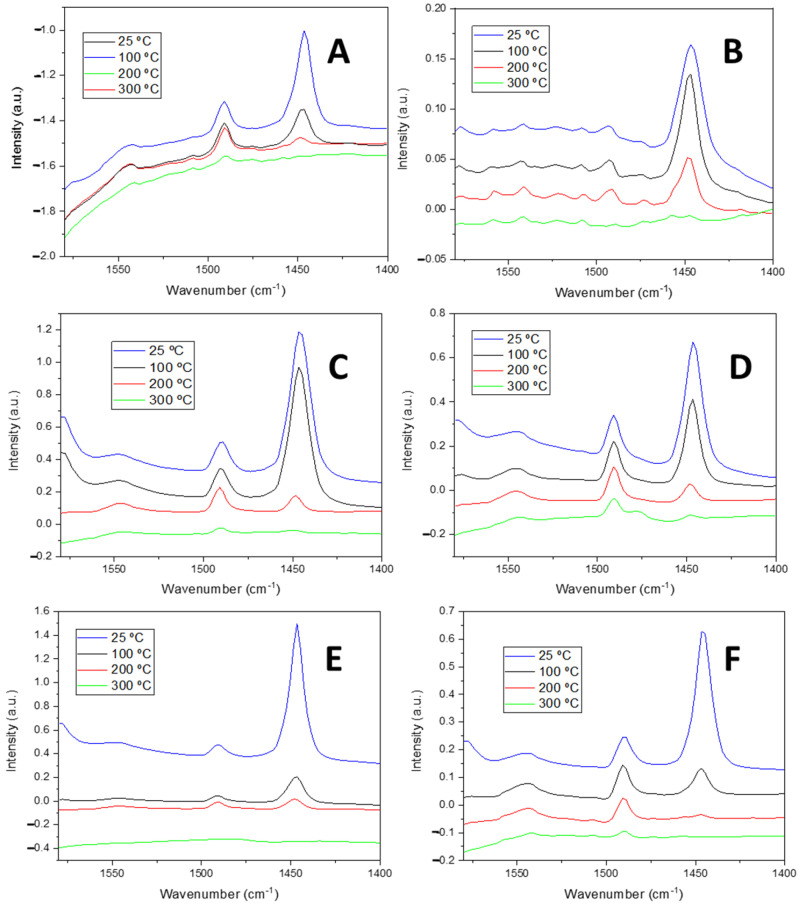
FTIR spectra after pyridine adsorption at 25 °C and desorption at different temperatures of (**A**) SBA-Nb, (**B**) SBA-Zr, (**C**) HMS-Nb, (**D**) SBA-NbZr, (**E**) HMS-NbZr, and (**F**) HMS-Zr.

**Figure 9 ijms-26-10054-f009:**
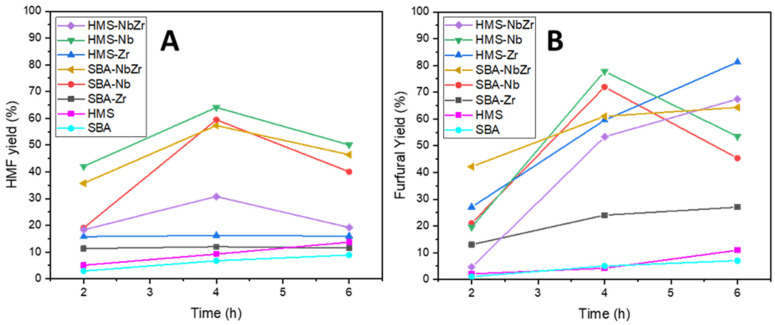
Performance of the catalysts for the production of (**A**) HMF at 180 °C and (**B**) furfural at 160 °C in an MIBK/water system (volume ratio = 3/1) and a catalyst/sugar weight ratio of 1/1.

**Figure 10 ijms-26-10054-f010:**
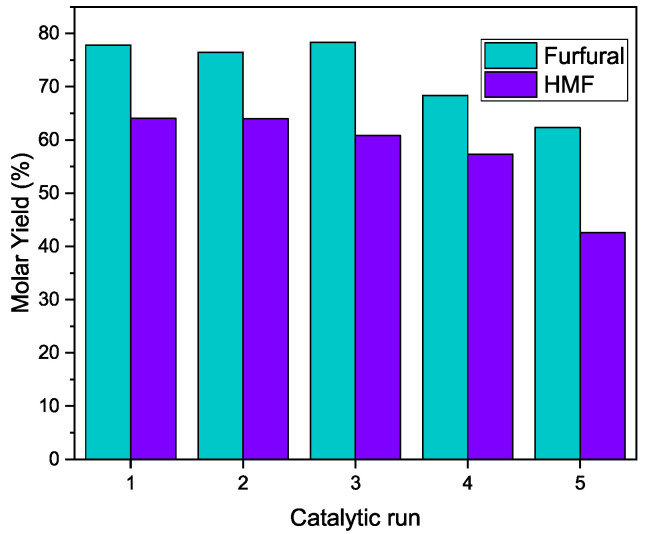
Reutilization of HMS-Nb for the production of furfural (160 °C, 4 h) and HMF (180 °C, 4 h).

**Figure 11 ijms-26-10054-f011:**
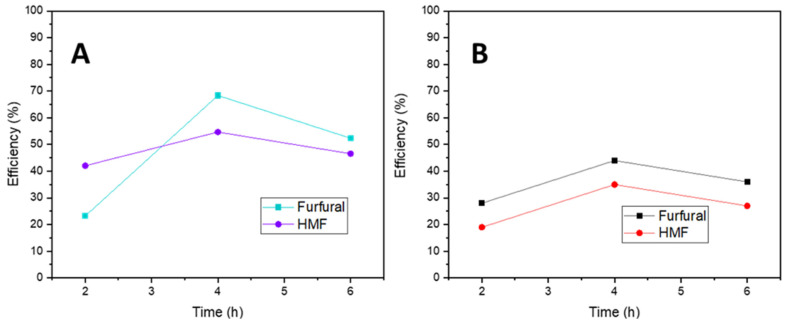
HMF and furfural production from *Ulva rigida* as rhamnose, xylose, and glucose feedstock using HMS-Nb catalyst (**A**) and Nb_2_O_5_ deposited over commercial fumed silica (**B**).

**Table 1 ijms-26-10054-t001:** Textural properties of the catalysts.

Catalysts	S_BET_ (m^2^/g)	t-Plot Micropore Area (m^2^/g)	External Surface Area (m^2^/g)	Pore Volume (cm^3^/g)	Average Pore Diameter (nm)
HMS	904	449	455	0.58	2.4
SBA	421	214	207	0.28	2.6
HMS-NbZr	988	855	132	0.54	3.7
HMS-Nb	1204	975	229	0.73	3.5
HMS-Zr	898	679	219	0.56	2.9
SBA-NbZr	424	113	310	0.30	3.6
SBA-Nb	555	150	405	0.42	3.6
SBA-Zr	445	241	204	0.23	3.4

**Table 2 ijms-26-10054-t002:** Atomic concentration of C, O, Si, Nb, and Zr on the surface of the catalysts.

Catalysts	C 1*s*	O 1*s*	Si 2*p*	Nb 3*d*	Zr 3*d*	Si/Nb	Si/Zr	Si/(Nb + Zr)
HMS-NbZr	9.49	63.39	25.49	7.24	5.32	3.52	4.79	2.03
HMS-Nb	8.40	64.09	27.23	4.80	--	5.67	--	--
HMS-Zr	10.16	63.18	25.90	--	3.77	--	6.87	--
SBA-NbZr	16.01	58.19	17.87	5.47	4.41	3.27	4.05	1.81
SBA-Nb	10.51	60.17	25.61	3.72	--	6.88	--	--
SBA-Zr	11.54	60.88	23.20	--	4.39	--	5.28	--

**Table 3 ijms-26-10054-t003:** Acid strength of the catalytic sites according to the NH_3_-TPD data.

Catalyst	Total Acidity (μmol/g)	Density of Acid Sites (μmol/m^2^)	Weak Sites ^a^ (μmol/g)	Moderate Sites ^b^ (μmol/g)	Strong Sites ^c^ (μmol/g)
HMS-NbZr	430	0.43	210	160	60
HMS-Nb	390	0.32	185	155	50
HMS-Zr	350	0.39	207	173	70
SBA-NbZr	460	1.08	293	107	60
SBA-Nb	360	0.65	191	97	72
SBA-Zr	220	0.49	136	54	30

^a^ 100–200 °C (weak acid sites); ^b^ 200–300 °C (moderate acid sites); ^c^ 300–550 °C (strong acid sites).

**Table 4 ijms-26-10054-t004:** Nature and strength of the acid sites of the catalysts according to pyridine adsorption coupled to FTIR.

Catalyst	Acid Nature	Weak Sites ^a^ (μmol/g)	Moderate Sites ^b^ (μmol/g)	Strong Sites ^c^ (μmol/g)
HMS-NbZr	Brønsted	24	13	22
	Lewis	82	14	7
HMS-Nb	Brønsted	10	24	11
	Lewis	147	31	20
HMS-Zr	Brønsted	30	26	29
	Lewis	64	6	0
SBA-NbZr	Brønsted	73	25	10
	Lewis	5	53	40
SBA-Nb	Brønsted	26	8	9
	Lewis	81	28	11
SBA-Zr	Brønsted	2	17	1
	Lewis	28	31	4

^a^ 100 °C; ^b^ 200 °C; ^c^ 300 °C.

## Data Availability

The original contributions presented in this study are included in the article/Appendix A. Further inquiries can be directed to the corresponding author.

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
