# Peer review of "Selective Dehydration of Pentoses and Hexoses of Ulva rigida to Platform Chemicals Using Nb2O5 and ZrO2 Supported on Mesoporous Silicas as Heterogeneous Catalysts"

_ijms, 2025, doi:10.3390/ijms262010054_

Round 1
Reviewer 1 Report
Comments and Suggestions for Authors
The present work reported the synthesis of Nb2O5 and ZrO2 supported on mesoporous silicas (HMS and SBA) as catalysts for the dehydration of monosaccharides contained in Ulva rigida biomass into furfural and 5-hydroxymethylfurfural. These prepared catalysts exhibited large surface areas, which promoted a high and stable catalytic performance.
Below are issues that need to be revised before proceeding to the following steps:
- In Section “Introduction”, a literature review of Nb and Zr-based nanomaterials developed for dehydration reactions should be additionally provided (paragraph 3).
- In Section 3.1.1, the surface area of pristine HMS and SBA should be measured and compared with that of prepared catalysts.
- XPS survey spectra of prepared samples should be provided in Supporting Information.
- In Section 3.1.4, TEM images of prepared samples did not clearly demonstrate the deposition of metal oxides over the support surfaces. For example, Figure 5B shows the typical structure of SBA without any indication of metal oxide particles or nanoparticles deposited. Similarly, Figure 5F shows a mixture of particles, but it is not evident which features correspond to the metal oxide nanoparticles, and which belong to the supports. The authors should clearly identify and distinguish the metal oxide particles from the support structures. TEM images of pristine HMS and SBA should be provided (preferably in Supporting Information) and compared with prepared catalysts to investigate if there were surface modifications after the metal oxide impregnation. Besides, the scale bars are too small.
- Results of the TEM analysis should be discussed in more detail.
- The discussion for the X-ray diffraction analysis should be put in the manuscript instead of Supporting Information.
- In Section 3.2, the catalytic performance of pristine HMS and SBA should also be investigated and compared with that of the prepared catalysts.
- In the Section Conclusion, the authors should provide more details of this work, including the results of material characterization.
Author Response
Thank you for giving us the opportunity to review our manuscript “Selective dehydration of pentoses and hexoses of Ulva rigida to platform chemicals using Nb2O5 and ZrO2 supported on mesoporous silicas as heterogeneous catalysts” to International Journal of Molecular Sciences (ijms-3875934).
To the best of our knowledge, we have addressed all the reviewers’ comments. We have uploaded our response to the reviewers, revised manuscript with changes highlighted in red.
We sincerely appreciate your time and effort, as well as those of the reviewers. We hope our revised manuscript is now appropriate for publication in International Journal of Molecular Sciences.
Sincerely,
Prof. Cristina García Sancho (on behalf of all authors)
Reviewer 2 Report
Comments and Suggestions for Authors
The study explores the catalytic conversion of sugars from Ulva rigida hydrolysates into furfural and HMF using monometallic and bimetallic Nbâ‚‚Oâ‚… and/or ZrOâ‚‚ supported on HMS and SBA silica. The catalysts showed good textural, acidic, and structural properties, and the authors achieved high molar yields with selective dehydration of pentoses and hexoses. HMS-Nb exhibited robust performance over multiple runs and maintained efficiency even with real algal liquor. While these results are promising, several limitations remain, and it is recommended that the authors address these issues before considering publication.
- In the abstract, the phrase “77.8% and 64.1% of of furfural and HMF molar yields” contains a redundant “of”; the extra “of” should be removed.
- In the sentence “The catalysts were obtained after calcination at 550 ºC , and presented”, the underline in 550 ºC should be deleted.
- It is recommended to add an “Entry” column on the leftmost side of each table and number each row accordingly. This will allow the authors to refer to specific rows in the main text simply by citing “entry XX,” making the discussion clearer and more concise.
- Overall, the figures in this manuscript are not sufficiently clear or aesthetically consistent. The authors are advised to standardize the formatting by enlarging and bolding the axis numbers and labels, as well as thickening the lines for better visibility.
- In Figure 2, the range from 0–60 should be enlarged to more clearly show the pore size distribution of the synthesized catalysts. Additionally, the authors should provide an explanation as to why the SBA-based catalysts presented a more heterogeneous pore size distribution, while the HMS catalysts showed a narrower distribution, rather than merely describing the phenomenon.
- In the sentence “data as no relevant crystallographic signals were observed (Figure S.I. 1)”, the reference Figure S.I. 1 should be corrected to Figure S1.
- In Figure 6, the vertical axis lacks scale marks, and the label should not be simply “signal”; it needs to be revised to provide a clear description of what is being measured.
- In Table 3, the superscripts “a, b, c” are indicated, but they are not explained or used in the table. The authors should clarify this.
- In Figure 7, the labels “A, B, C, D, E, F” are too large and should be resized appropriately.
- Although the authors mention a footnote labeled “c” in the table caption, Table 4 contains two footnotes labeled “b” and no footnote “c.” This inconsistency should be corrected.
- The references should be formatted according to the requirements of the IJMS. In addition, many journal names are not correctly abbreviated and should be revised accordingly.
Author Response

(The authors gave the same response as above.)

Round 2
Reviewer 1 Report
Comments and Suggestions for Authors
The authors have provided a thorough review. I recommend publication of this work in its current form.
Reviewer 2 Report
Comments and Suggestions for Authors
In my opinion, the authors have adequately addressed the required corrections and revisions, and therefore the manuscript is suitable for publication. However, regarding Comment 8, the superscripts a, b, and c should be added directly in Table 3, rather than only being included in the table footnote.